# An optimal growth law for RNA composition and its partial implementation through ribosomal and tRNA gene locations in bacterial genomes

Xiao-Pan Hu🄳, Martin J. Lercher🄳*

Institute for Computer Science and Department of Biology, Heinrich Heine University, Düsseldorf, Germany

* martin.lercher@hhu.de

## Abstract

The distribution of cellular resources across bacterial proteins has been quantified through phenomenological growth laws. Here, we describe a complementary bacterial growth law for RNA composition, emerging from optimal cellular resource allocation into ribosomes and ternary complexes. The predicted decline of the tRNA/rRNA ratio with growth rate agrees quantitatively with experimental data. Its regulation appears to be implemented in part through chromosomal localization, as rRNA genes are typically closer to the origin of replication than tRNA genes and thus have increasingly higher gene dosage at faster growth. At the highest growth rates in *E. coli*, the tRNA/rRNA gene dosage ratio based on chromosomal positions is almost identical to the observed and theoretically optimal tRNA/rRNA expression ratio, indicating that the chromosomal arrangement has evolved to favor maximal transcription of both types of genes at this condition.

## Author summary

Unlike the proteome composition, RNA composition is often assumed to be independent of growth rate in bacteria, despite experimental evidence for a growth rate dependence in many microbes. In this work, we derived a growth-rate dependent optimal tRNA/rRNA concentration ratio by minimizing the combined costs of ribosome and ternary complex at the required protein production rate. The predicted optimal tRNA/rRNA expression ratio, which is a monotonically decreasing function of growth rate, agrees with experimental data for *E. coli* and other fast-growing microbes. This indicates the existing of an RNA composition growth law. Due to the presence of partially replicated chromosomes, gene dosage is higher for those genes whose DNA is replicated earlier, an effect that becomes stronger at higher growth rates. Because rRNA genes are located closer to origin of replication than tRNA genes in fast-growing species, the tRNA/rRNA gene dosage ratio scales with growth rate in the same direction as the optimal tRNA/rRNA expression ratio. Thus, it appears that the RNA growth law is–at least in part–implemented simply through the genomic positions of tRNA and rRNA genes. This finding indicates that growth rate-dependent optimal resource allocation can influence the genomic organization in bacteria.

**Data Availability Statement:** All data used in the manuscript is provided in S1–S5 Tables.

**Funding:** This work was supported by grants to M. J.L. from the Volkswagen Foundation under the

"Life?" initiative and from the German Research Foundation (DFG) through CRC 1310, and, under Germany's Excellence Strategy, through EXC 2048/1 (Project ID: 390686111). The funders had no role in study design, data collection and analysis, decision to publish, or preparation of the manuscript.

**Competing interests:** The authors have declared that no competing interests exist.

## Introduction

The systematic change of the coarse-grained composition of bacterial proteomes with growth rate [1,2] can be quantified through phenomenological growth laws [3,4]. The most prominent growth law describes an apparently linear increase of the ribosomal protein fraction with growth rate [1,3]. These laws have been successfully applied to the prediction of a range of phenotypic observations [3,5–8]. Recently, it has been argued that they arise from an optimal balance between the cellular investment into catalytic proteins and their substrates [9].

In contrast to the proteome composition, the partitioning of bacterial RNA into messenger (mRNA), ribosomal (rRNA), and transfer (tRNA) RNA is often assumed to be growth rate-independent [2,3,5,6,10,11]. For example, the assumption of a constant RNA composition has been used to estimate an empirical relationship for the macromolecular cellular composition across bacterial species [12,13]. However, experimental evidence from multiple species suggests that the tRNA/rRNA expression ratio decreases monotonically with growth rate [14–22], suggesting the existence of a bacterial growth law for RNA composition.

The regulatory implementation of bacterial growth laws is generally assumed to arise from a small number of major transcriptional regulators such as ppGpp [23,24] and cAMP [4,25]. However, growth-rate dependent transcriptional regulation could also be implemented through chromosomal gene positioning. In many prokaryotes, the cellular doubling time can be even shorter than the time required for genome replication. To coordinate DNA replication and cell division, fast-growing prokaryotes re-initiate DNA replication before the previous round of replication is complete. In this case, genes closer to oriC have more DNA copies than genes further away in the genome, a phenomenon described as replication-associated gene dosage effects (below, we use "gene dosage" to refer to the growth rate-dependent average DNA copy number per cell of a given gene). Prokaryotic genes are non-randomly located on multiple levels [26–28], with highly expressed genes biased towards the origin of replication (oriC) [29]. The latter observation is thought to facilitate high expression levels at fast growth due to replication-associated gene dosage effects [30–32]. Indeed, chromosome rearrangements that shift highly expressed genes from the origin to the terminus of replication reduce fitness [33–37].

rRNA forms the central part of the catalyst of peptide elongation, while tRNA forms the core of the substrate; together, they account for the bulk of cellular RNA [2]. Their cytosolic concentrations at different growth rates in *E. coli* are well described by an optimality assumption [9,38,39]. Moreover, chromosomal gene positions in *E. coli* are known to affect the expression of both tRNA and rRNA genes [40,41]; both types of genes are located closer to oriC in fast- compared to slow-growing bacteria, with rRNA genes positioned closer to oriC than tRNA genes in most examined fast-growing bacteria [29].

Based on these previous observations, we hypothesize (i) that the relative expression of tRNA and rRNA can be described by a bacterial growth law that arises from optimal resource allocation and (ii) that this growth law is at least partially implemented through the relative chromosomal positioning of tRNA and rRNA genes.

## Results and discussion

### An RNA growth law resulting from maximal efficiency of translation

Cellular dry mass density appears to be approximately constant across conditions [42,43]. Dry mass may thus be considered a limiting resource [9,39] if the dry mass density is occupied by one particular molecule, less will be available for all other molecules. In terms of dry mass allocation, translation is the most expensive process in fast-growing bacteria [2,44]. Thus, at a given protein synthesis rate, it is likely that natural selection will act to minimize the summed

dry mass density of all translational components. As evidenced by comparison of diverse data to a detailed biochemical model of translation, the allocation of cellular resources across components of the *E. coli* translation system minimizes their total dry mass concentration at a given protein production rate [39]. This result indicates that natural selection indeed favored the parsimonious allocation of cellular resources to the translation machinery in *E. coli*.

To generalize this optimization hypothesis to other species, we here analyze a coarse-grained translation model that only considers peptide elongation, where the active ribosome acts as an enzyme that converts ternary complexes (TC), consisting of elongation factor Tu (EF-Tu), GTP, and charged tRNA, into an elongating peptide chain following Michaelis-Menten kinetics (Fig 1A) [5,45]. In exponential, balanced growth at rate $\mu$ with cellular protein

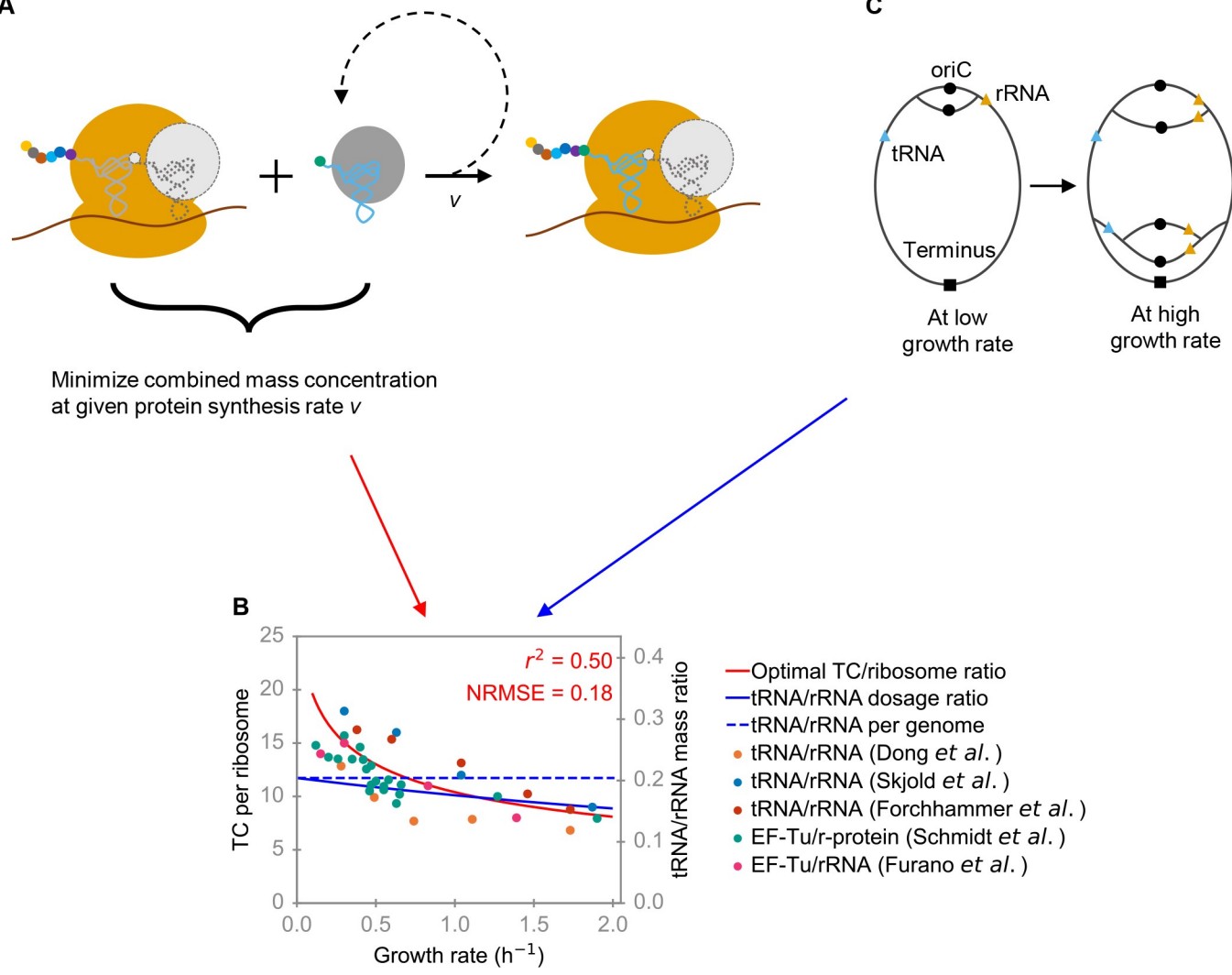

**Fig 1. The RNA growth law and its implementation through gene positions.** (A) Coarse-grained protein translation model, following Michaelis-Menten kinetics with the active ribosome as catalyst and TC as substrate. The optimal TC/ribosome expression ratio is derived by minimizing the combined mass concentration of ribosome and TC at the given protein synthesis rate $\nu$. (B) Different experimental estimates of TC/ribosome expression ratios in *E. coli* (points, colors indicate the data source) are consistent with the optimal ratio according to Eq (1) (red line) (Pearson's $r^2 = 0.50$; NRMSE = 0.18). The dashed blue line indicates the genomic tRNA/rRNA ratio, the solid blue line indicates the tRNA/rRNA gene dosage ratio estimated from Eq (21). (C) A schematic diagram showing the dosage ratio of two genes as a function of growth rate. If rRNA genes are located on average closer to oriC than tRNA genes–which is the case in *E. coli*–then the dosage of rRNA genes will increase faster with increasing growth rate than that of the tRNA genes; consequently, the tRNA/rRNA gene dosage ratio becomes a decreasing function of growth rate (solid blue curve in panel B).

concentration $[P]$, the total rate of protein production is $v = \mu \cdot [P]$. We derived the optimal concentration ratio between TC (with molecular mass $m_{TC}$) and ribosome ($R$, with molecular mass $m_R$) at this production rate by minimizing their combined mass concentration, $M_{total} = m_{TC}[TC] + m_R[R]$ (Methods):

$$\frac{[TC]}{[R]} = \frac{a \cdot k_{cat}}{\sqrt{a \cdot \mu \cdot [P] \cdot k_{on}^{diff} + k_{cat}}};$$

(1)

here, $a = \frac{m_R}{m_{TC}} = 33.1$ is the ratio of molecular weights of ribosome and TC; $k_{cat}$ is the turnover number of the ribosome; and $k_{on}^{diff}$ is the diffusion-limited binding constant of TC to ribosome [5], which can be treated as a constant if the cell density is approximately constant across species.

For a given genome, $a$ and $k_{cat}$ are constants [5,6]. Moreover, the cellular protein concentration $[P]$ (in terms of amino acid residues) appears to be similar across most species [46] and shows only minor variations across growth rates in those bacteria where it has been tested [7,47,48]. Thus, Eq (1) predicts that in any given species, the TC/ribosome expression ratio is a monotonically decreasing function of the growth rate $\mu$. Since most cellular EF-Tu and tRNA are present in the form of TCs [5], hereafter, the TC concentration is assumed to be approximately equal to the concentrations of EF-Tu and tRNA.

To calculate the optimal TC/ribosome expression ratio in *E. coli*, we use the measured protein concentration $[P]$ [49], set the turnover number $k_{cat}$ to the maximal observed translation rate [2], and set $k_{on}^{diff}$ to the diffusion limit of the TC [5] (Methods; see also Ref. [39]). Fig 1B compares the optimal predictions (red line) to experimental datasets for *E. coli* that estimated the TC/ribosome expression ratio based on ratios of tRNA/rRNA [22,50,51], EF-Tu/rRNA [21], and EF-Tu/ribosomal proteins [49] (S1 Table). The Pearson correlation between observed and fitted data is $r^2 = 0.50$, $P = 5.9 \times 10^{-7}$ (root-mean-square error normalized by observed mean, NRMSE = 0.18); these measures have to be interpreted against the variability between the diverse datasets. Consistent with the predictions, all experimental estimates of the TC/ribosome expression ratio are approximately two-fold higher at low compared to high growth rates. As the TC and ribosome constitute the two major components of cellular RNA [2], we conclude that the optimal TC/ribosome expression ratio according to Eq (1) represents a bacterial growth law for RNA composition:

$$\frac{M_{tRNA}}{M_{rRNA}} = r \cdot \frac{k_{cat}}{\sqrt{a \cdot \mu \cdot [P] \cdot k_{on}^{diff} + k_{cat}}}$$

(2)

where $M_{tRNA}$ and $M_{rRNA}$ are the cellular mass of tRNA and rRNA, respectively, and $r = 0.58$ is the ratio of the tRNA mass fraction of a TC and the rRNA mass fraction of the bacterial ribosome (Methods).

The proteome degradation rate in *E. coli* is typically 0.02–0.04 h$^{-1}$ [52–54], which is much smaller than the maximal growth rate. Accordingly, including protein degradation into the model only affects the predictions at very low growth rates in *E. coli* (Fig A in S1 File). In contrast, protein degradation may have a large impact on the RNA growth law for species with degradation rates comparable to their maximal growth rates. Further, while our model assumes that all tRNA and ribosome are active, there is evidence for a substantial fraction of de-activated ribosomes and TCs at low growth rates in *E. coli* [39]. This approximation may contribute to the discrepancy between our predictions and data at low growth rates.

In previous work by Klumpp *et. al.*, the optimal TC/ribosome expression ratio was predicted by considering protein mass instead of dry mass as the limiting resource [5]; these authors identified the proteome fractions allocated to ribosomes and TCs that maximize

growth rate in a very similar model of protein translation to that used here. This optimal proteome allocation results in a substantial lower predicted TC/ribosome expression ratio compared to the experimentally observed data (Fig B in S1 File). Our hypothesis of parsimonious dry mass allocation, which considers RNA and protein masses equally, explains the observed TC/ribosome expression ratio much better than optimal proteome allocation alone.

## The RNA growth law is partially implemented through genomic positions in *E. coli*

Above, we have shown the existence of an RNA growth law in *E. coli*, reflecting a decrease of the optimal tRNA/ribosome expression ratio with increasing growth rate. Given that the genomic position of rRNA genes is typically closer to oriC than that of tRNA genes in bacteria [29], we hypothesize that this growth rate-dependence may–at least in part–be implemented through replication-associated gene dosage effects.

To test our hypothesis, we used the model developed by Bremer and Churchward [32] to quantify the dosage ratio of two genes at growth rate $\mu$,

$$\frac{\overline{X_i}}{\overline{X_j}} = e^{C \cdot \mu \cdot (position_j - position_i)} \tag{3}$$

here, for gene $i$, $\overline{X_i}$ is the dosage and $position_i$ is the position; $C$ is the time required to complete one round of chromosome replication (see Methods for details, and see Text A in S1 File for the effect of a growth rate-dependent C period on the dosage ratio for tRNA and rRNA genes). Clearly, the dosage ratio of two genes with different chromosomal positions is a monotonous function of $\mu$. As shown schematically in Fig 1C, if a rRNA gene is located closer to oriC than a tRNA gene, the tRNA/rRNA gene dosage ratio (reflecting chromosomal copy numbers) will be a decreasing function of growth rate, just as the optimal tRNA/rRNA expression ratio (reflecting RNA production; Fig 1B).

Consistent with a (partial) implementation of the RNA growth law through genomic positioning, the rRNA genes are, on average, located closer to oriC than tRNA genes in *E. coli*, with genomic position 0.20 ± 0.17 (mean ± standard deviation) for rRNA genes and 0.45 ± 0.27 for tRNA genes (see Fig C in S1 File for the distributions). The difference in genomic positions between tRNA and rRNA genes results in a growth rate-dependent tRNA/rRNA gene dosage ratio (solid blue curve in Fig 1B) that agrees qualitatively with the optimality predictions from Eq (2) (to calculate the dosage ratio across multiple genes, we used Eq (21), a generalized version of Eq (3), see Methods). For comparison, Fig 1B also shows the constant tRNA/rRNA genomic ratio, i.e., the ratio of gene copy numbers per complete chromosome (dashed blue line).

As all necessary parameters are available for *E. coli*, we can make quantitative predictions for the tRNA/rRNA expression ratio without adjustable parameters. It is notable that according to Fig 1B, the tRNA/rRNA gene dosage ratio at high growth rates ($1\,h^{-1} \leq \mu \leq 2\,h^{-1}$) is very close to the optimal tRNA/rRNA expression ratio, which corresponds to about 9 tRNAs per ribosome (Fig 1B). This result is consistent with the notion that at the highest growth rates, both tRNA and rRNA genes are transcribed at the maximal possible rate, such that their relative expression is dominated by gene dosage effects in these conditions. The expression of both tRNA and rRNA operons is regulated by the P1 promoter, which is repressed by ppGpp; at near-maximal growth rates, ppGpp concentrations are low, and the P1 promoter works near its maximal capacity [55]. In contrast, at low growth rates, P1 is repressed by ppGpp, and thus gene dosage can only partially explain the tRNA/rRNA expression ratio in these conditions.

### The RNA growth law in fast-growing microbes beyond *E. coli*

The approximate Michaelis-Menten form of the rate law for peptide elongation, on which the RNA composition growth law is based, arises from the structure of the detailed elongation process [45]. As this process is shared by all living cells [45], we expect that the RNA composition growth law, Eq (2), also holds for other fast-growing microbes (with $a$ = 40.3 and $r$ = 0.59 in eukaryotes, Methods). To test this hypothesis, we collected all available tRNA/rRNA expression ratios in microbes (Fig 2A and S2 Table). Note that if protein concentration [$P$] and the cellular dry mass density are indeed approximately constant across species [46], then Eqs (1) and (2) contain a single species-specific parameter, $k_{cat}$.

For six out of the seven datasets in Figs 1B and 2A, the tRNA/ribosome expression ratio decreases with increasing growth rate. The only exception, the cyanobacterium *Synechococcus elongatus*, has a much smaller maximal growth rate ($\mu_{max}$ = 0.23 h$^{-1}$) than the other species, and its tRNA/rRNA expression ratio does not show a clear growth rate-dependence (Fig 2A, Spearman's $\rho$ = -0.01, $P$ = 0.98) [56]. It is conceivable that slow-growing species do not fully optimize their translation machinery composition, as a near-optimal constant TC/ribosome expression ratio may incur a lower fitness cost than the expression of a regulatory system for growth rate-dependent optimal expression.

To verify the implementation of the RNA growth law in the remaining, fast-growing species, we used our model to estimate $k_{cat}$ by fitting the measured tRNA/rRNA expression ratio to Eq (1) (solid lines in Fig 2A). Independently, we also estimated the effective ribosome

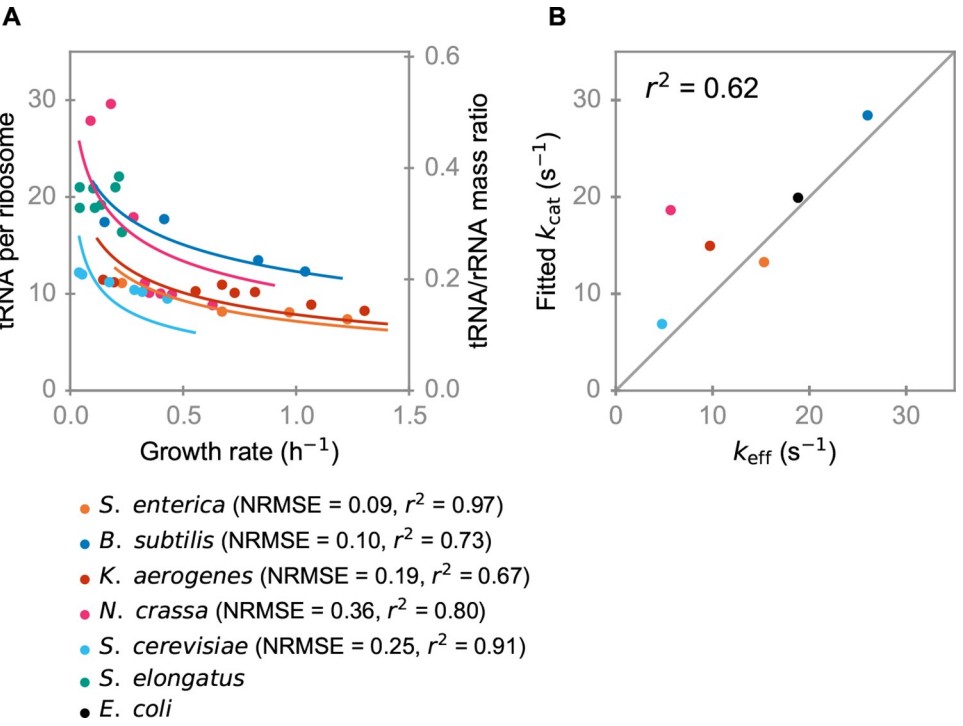

**Fig 2. The RNA growth law across species.** (A) Experimentally observed tRNA/ribosome expression ratios in different microbes decrease with growth rate, consistent with the predicted optimal tRNA/ribosome expression ratio. For each species except *S. elongatus*, which is a slow-growing species and shows no systematic growth rate dependence, we fitted Eq (1) to the data by varying the single adjustable parameter $k_{cat}$ (solid lines; the numbers in parentheses after the species names quantify the agreement between the fitted lines and the data). Note that the y-axis on the right-hand side is based on the tRNA/rRNA mass ratio for bacteria. For eukaryotic microbes, the tRNA/rRNA mass ratio should be scaled by a factor of 0.84 according to Eq (15). (B) Comparison of fitted $k_{cat}$ and effective ribosome turnover number $k_{eff}$.

turnover number ($k_{\text{eff}}$) through the relationship $\mu \cdot [P] = k_{\text{eff}} \cdot [R]$, using measured values for $\mu$, $[P]$, and $[R]$ (S3 Table; fitting was performed for all species excluding *S. elongatus*, in which the tRNA/rRNA expression ratio is independent of the growth rate and thus a fitting procedure would be meaningless). Fig 2B shows a close correspondence between the $k_{\text{cat}}$ values estimated via Eq (1) and the effective turnover numbers (Pearson's $r^2 = 0.62$, $P = 0.063$). Given that the tRNA/rRNA expression ratios used for fitting Eq (1) were measured with different experimental methodologies by different groups, we do not expect a perfect correlation; that our model still explains 62% of the variation appears to strongly support our analyses. We thus conclude that Eq (2) describes a universal RNA growth law for fast-growing bacterial species.

## Implementation of the RNA growth law through tRNA and rRNA genomic positions across bacteria

Next, we asked if other bacteria also show a differential distribution of tRNA and rRNA genes along the chromosome that is consistent with a partial implementation of the RNA growth law through replication-associated gene dosage effects. As a strong selection pressure toward optimal tRNA/ribosome expression ratios is expected mainly in fast-growing species (Fig 2A), we surveyed gene positions in bacteria for which maximal growth rates are available [57]. In *E. coli*, the summed time of DNA replication (C period, ~ 40 min) and cell division (D period, ~20 min) [31] is approximately 1 h. Given that these times will be roughly similar in many other species, we assume that species with substantially larger doubling times are unlikely to perform multiple simultaneous rounds of replication, while cells with shorter doubling times will likely perform multiple replication rounds simultaneously and hence experience stronger replication-associated gene dosage effects. Accordingly, we classified bacteria with doubling times ≤1 h (i.e., $\mu_{\text{max}} \geq 0.69$ h⁻¹) as fast-growing species, and bacteria with doubling times > 1 h as slow-growing species.

As shown in Fig 3A and 3B (orange points), we found that in fast-growing species, rRNA and tRNA genes are generally located in the vicinity of oriC, at relative positions < 0.5 (0.5 is located 0.25 genome lengths to either side of oriC, halfway between oriC and the terminus of replication; for each genome represented in Fig 3, the positions are the arithmetic means across the corresponding genes). This observation is consistent with previous analyses [29,57]. Moreover, we found that both rRNA and tRNA genes tend to be located ever closer to oriC with increasing $\mu_{\text{max}}$ (correlation with $\mu_{\text{max}}$ for *position*$_{\text{rRNA}}$: Spearman's $\rho = -0.59$, $P = 9.2 \times 10^{-6}$, $P$-value calculated based on phylogenetically independent contrasts [58] to control for phylogenetic non-independence between datapoints: $P_{\text{ic}} = 0.04$; for *position*$_{\text{tRNA}}$: $\rho = -0.40$, $P = 4.6 \times 10^{-3}$, $P_{\text{ic}} = 2.1 \times 10^{-4}$). In slow-growing species, rRNA genes still tend to be close to oriC (Fig 3A, blue points; one sample Wilcoxon signed rank test, $P = 2.8 \times 10^{-10}$), while tRNA genes are distributed around the midpoint between oriC and the terminus (Fig 3B, blue points; one sample Wilcoxon signed rank test, $P = 0.11$).

As expected from our hypothesis of a partial implementation of the RNA growth law through replication-associated gene dosage effects, we found that rRNA genes are closer to oriC than tRNA genes in most slow-growing and in all but one fast-growing bacteria (Fig 3C; note that the one exception has a small genome of only 1.8 Mb). Accordingly, the tRNA/rRNA expression ratio that would be obtained if regulation was exclusively through gene dosage would be a decreasing function of growth rate, in qualitative agreement with the optimality predictions from Eq (2). This finding, together with our detailed analysis of individual species (Figs 1B and 2), supports our hypothesis that natural selection has fine-tuned the positions of tRNA and rRNA genes to match the RNA growth law for optimally efficient translation in fast-growing species.

The maximal growth rate $\mu_{\text{max}}$ is not the only factor that affects the strength of replication-associated gene dosage effects. At the same DNA replication rate, smaller genomes need less

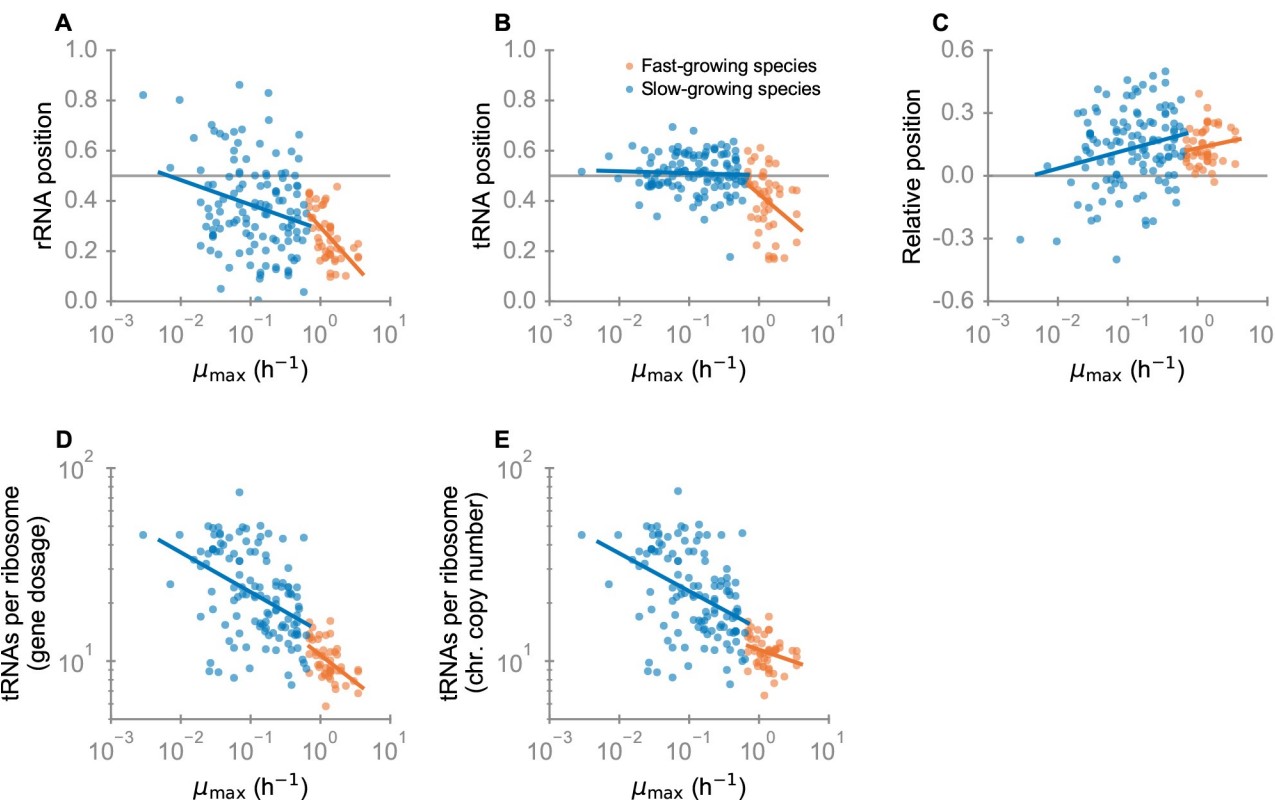

**Fig 3. The genomic positions of rRNA and tRNA genes implement the RNA growth law in fast-growing species.** (A) Arithmetic means of the rRNA positions for individual genomes as a function of $\mu_{max}$. The horizontal grey line (position 0.5) marks the midpoint between origin and terminus of replication. (B) Same for tRNA. (C) Relative positions between tRNA and rRNA genes ($position_{tRNA}$—$position_{rRNA}$). (D) tRNA/rRNA gene dosage ratios. (E) Genomic tRNA/rRNA ratios (per chromosome). Blue points indicate slow growing species (with blue linear regression line), orange points indicate fast-growing species (with orange linear regression line).

time to replicate than larger genomes. Thus, at the same growth rate, bacteria with smaller genomes are expected to have fewer replication forks in the cell, and hence experience weaker gene dosage effects. Text B in S1 File explores the influence of genome size on the positioning of tRNA and rRNA genes; here, we only provide a brief summary. Consistent with the above notions, in fast-growing species, we found that the position of rRNA genes is negatively correlated with genome size, i.e., there appears to be less selection pressure toward positioning rRNA genes close to oriC in smaller genomes. At the same time, the relative genomic position of tRNA and rRNA genes is positively correlated with genome size in fast-growing species, again indicating lower selection pressures toward specific genomic positions is smaller genomes. However, in a combined statistical model, $\mu_{max}$ remains the main predictor of tRNA and of rRNA positions in fast-growing species, with only marginal contributions from genome size. It is conceivable that the effective population size–which influences the efficiency of natural selection–also influences the genomic positions of tRNA and rRNA genes. However, we found no evidence for such an influence (Fig D in S1 File).

For the multi-species dataset, we have so far only considered the genomic positions. We now turn our attention to the resulting tRNA/rRNA gene dosage ratio at the reported maximal growth rate. According to Eq (1), faster growing species need a lower TC/ribosome expression ratio at maximal growth. We indeed find statistically highly significant negative correlations between the predicted tRNA/ribosome gene dosage ratio (Eq (21)) and $\mu_{max}$ (Fig 3D; slowly growing species: $\rho = -0.44$, $P = 2.8 \times 10^{-7}$, $P_{ic} = 6.0 \times 10^{-4}$; fast-growing species: $\rho = -0.49$,

$P = 4.3 \times 10^{-4}$, $P_{ic} = 0.037$) (see Text C in S1 File for the treatment of tRNA genes; these calculations assume a constant DNA replication rate $k_{rep} = 1000$ s$^{-1}$ across species, see Text A in S1 File for species-specific replication rate $k_{rep}$).

While slowly growing species show a wide range of tRNA/ribosome gene dosage ratios, the ratio in fast-growing species shows a much tighter distribution (*F*-test for equality of variances: $P < 10^{-15}$). In slow-growing species, the effects of replication-associated gene dosage effects are weak: the tRNA/ribosome gene dosage ratios are almost identical to the corresponding chromosomal copy number ratios (Fig 3E). In fast-growing species, the chromosomal tRNA/rRNA gene copy number ratios show a distribution that is similarly tight as that for the corresponding gene dosage ratios (*F*-test for equality of variances: $P < 10^{-15}$). As expected, species harbor increasingly more tRNAs and ribosomal genes with increasing $\mu_{max}$; consistent with the RNA growth law, this effect also leads to a negative correlation between the number of tRNA genes and the tRNA/ribosome (gene dosage and genomic) ratios (Fig E in S1 File): at higher maximal growth rates, bacteria have more tRNA genes, but the number of ribosomal genes increases even faster. In contrast to the rRNA and tRNA gene positions (Fig 3A and 3B) and the gene dosage ratios (Fig 3D), the tRNA/rRNA chromosomal copy number ratios show no strong systematic dependence on $\mu_{max}$ in fast-growing species (Fig 3E, $\rho = -0.24$, $P = 0.10$, $P_{ic} = 0.36$). Interestingly, we also find no statistically significant dependence of the relative position (*position*$_{tRNA}$—*position*$_{rRNA}$) on $\mu_{max}$ in fast-growing species (Fig 3C, $\rho = 0.15$, $P = 0.31$, $P_{ic} = 0.15$).

All these findings indicate that in fast-growing species, not only the absolute numbers of rRNA and tRNA genes, but also the relative numbers of tRNA and rRNA genes (tRNA/rRNA gene dosage ratio and tRNA/rRNA genomic ratio) are tightly constrained, consistent with the optimization of the translation machinery composition according to the RNA growth law and its implementation through replication-associated gene dosage effects.

## Impact of the RNA growth law on cell growth and genome organization

Above, we describe and explain a systematic dependence of RNA composition on growth rate in fast-growing bacteria. Why then does the assumption of a growth rate-independent RNA composition work well in theoretical models for the growth of *E. coli* under various perturbations [3,6,10,11]? We derived the RNA growth law from an assumption of parsimonious dry mass utilization by the protein translation machinery, in our simple model represented by TCs and ribosomes. As detailed in Text D in S1 File, we find that at intermediate to high growth rates in *E. coli*, the optimal combined mass concentration of ribosomes and TCs is very similar to the combined mass concentration under the assumption of a constant tRNA/rRNA expression ratio, with a 4.4% difference at $\mu = 0.2$ h$^{-1}$ and much smaller differences at higher growth rates (Fig K in S1 File). Thus, except at the lowest growth rates, the optimal RNA composition will only have a small impact on predictions of cellular growth rates. However, even growth rate differences on the order of 1% or less are highly relevant in evolutionary terms for natural bacterial populations, explaining why we find systematic evidence for the optimal expression of ribosomes and TCs (Figs 1B and 2A) and the differential genomic positions of rRNA and tRNA genes (Fig 3A–3C) across bacterial species.

## Model limitations

The derivation of the RNA growth law, Eq (2), is based on a coarse-grained protein translation model, where the ribosome acts as a catalyst that consumes TCs according to irreversible Michaelis-Menten kinetics. This coarse-grained model ignores many details of the molecular processes contributing to protein translation, such as the rate parameters for individual subprocesses [59,60] and the occurrence of traffic jams of ribosomes co-translating the same mRNA [61]. Following earlier work [5], we absorb the effects of these detailed processes on the

translation rate into the effective ribosomal turnover number, $k_{cat}$, which we treat as a species-specific constant. The agreement between the predictions derived from the coarse-grained model and experimental data (in particular Figs 1B and 2B) indicate that these simplifications represent an appropriate approximation.

One important parameter not explicitly considered here is temperature. At cold stress, the DNA replication rate becomes much slower in *E. coli* [62]. Experimental data shows that at low temperatures, the gene dosage ratio is almost constant across growth rates in *E. coli* (Fig F in S1 File). In our analyses, we only considered species-specific optimal growth temperatures, appropriate for the experimental data underlying Figs 1 and 2, and for the maximal growth rates considered in Fig 3. It appears not unlikely that the fine-tuned coordination between tRNA and ribosome expression breaks down at temperatures far away from optimal growth conditions.

Moreover, we here consider only the average genomic positions of tRNA and rRNA genes. While the optimal scaling of the tRNA/rRNA expression ratio (Eq (2)) with growth rate is independent of codon frequencies, it is still conceivable that selection pressure toward specific genomic positions is stronger for tRNA genes whose products decode more abundant codons. However, we found no such systematic dependence across genomes (Text C in S1 File).

## Conclusion

In sum, the tRNA/ribosome expression ratio appears to be tightly constrained across fast-growing bacteria. At fast growth, its regulation is likely dominated by replication-associated gene dosage effects, implemented through the relative chromosomal positioning of tRNA and ribosomal RNA genes. The objective of this regulation is to not only increase the expression of TCs and ribosomes with growth rate, but to also adjust their relative concentrations according to the RNA composition growth law quantified by Eqs (1) and (2).

## Methods

### Derivation of the optimal TC/ribosome expression ratio

In recent work, we have shown that the growth-rate dependent composition of the translation machinery in *E. coli* is accurately described by predictions based on detailed reaction kinetics and the numerical minimization of the total mass of all participating molecules [39]. This minimization was motivated by the observation that the cellular dry mass density is approximately constant across growth conditions [42]. Accordingly, if part of the dry mass density is occupied by one particular molecule type, less will be available for all other molecule types. This reasoning assumes that cellular dry mass is a growth-limiting resource; considering other growth-limiting resources, such as the minimization of the energy consumed or the enzyme mass required for the production of the different molecules led to almost identical results [39].

Here, we consider a much simpler representation of the elongation step of protein translation, which can be modeled as an enzymatic reaction following Michaelis-Menten kinetics [5]. In this case, the minimization of the combined mass concentration of ribosome and TC can be performed analytically, as demonstrated by Dourado *et al.* [9]; following this work, we here briefly summarized the derivation of the optimal TC/ribosome expression ratio.

In the coarse-grained protein translation model [5], the protein synthesis rate *v* can be expressed as

$$v = k_{cat}[R] \frac{[TC]}{K_m + [TC]} \tag{4}$$

Here, $k_{cat}$ is the effective turnover number of the ribosome, and $K_m$ is the ribosome's Michaelis constant for TC. The combined cytosol mass density of ribosome and TC is given by

$$c = [R] \cdot m_R + [TC] \cdot m_{TC}, \tag{5}$$

where $m_R$ is the molecular weight of the ribosome, and $m_{TC}$ is the molecular weight of the TC. We can express the ribosome concentration $[R]$ as a function of $v$ by rearranging Eq (4),

$$[R] = \frac{v}{k_{cat}} \left( \frac{K_m}{[TC]} + 1 \right) \tag{6}$$

Substituting Eq (6) into Eq (5), we have

$$c = \frac{v}{k_{cat}} \left( \frac{K_m}{[TC]} + 1 \right) \cdot m_R + [TC] \cdot m_{TC} \tag{7}$$

At a given protein production rate $v$, $c$ is now only a function of the TC concentration. The minimal $c$ can then be obtained by setting the derivative of Eq (7) with respect to $[TC]$ to zero,

$$\frac{dc}{d[TC]} = m_{TC} - \frac{m_R K_m v}{k_{cat}} \frac{1}{[TC]^2} = 0 \tag{8}$$

With the ribosome/TC mass ratio $a = m_R/m_{TC}$, the optimal $[TC]$ can be expressed as

$$[TC] = \sqrt{\frac{m_R K_m v}{m_{TC} k_{cat}}} = \sqrt{\frac{a K_m v}{k_{cat}}} \tag{9}$$

Substituting Eq (9) into Eq (3), the optimal ribosome concentration $[R]$ can be expressed as

$$[R] = \frac{v}{k_{cat}} + \sqrt{\frac{K_m v}{a k_{cat}}} \tag{10}$$

Thus, the TC/ribosome concentration ratio can be written as

$$\frac{[TC]}{[R]} = \frac{a \cdot \sqrt{k_{cat} \cdot K_m}}{\sqrt{a \cdot v} + \sqrt{k_{cat} \cdot K_m}} \tag{11}$$

At steady state, the protein production rate $v$ is equal to rate of protein dilution by volume growth,

$$v = \mu \cdot [P], \tag{12}$$

with growth rate $\mu$ and total cellular protein concentration $[P]$ (in units of amino acids per volume).

As the binding between the ribosome and the TC is limited by the diffusion of the TC, $K_m$ can be approximated through $K_m \approx k_{cat}/k_{on}^{diff}$, with $k_{on}^{diff}$ the diffusion-limited binding constant of the TC to the ribosome [5]. Thus, Eq (11) can be rewritten as (Eq (1) of the main text)

$$\frac{[TC]}{[R]} = \frac{a \cdot k_{cat}}{\sqrt{a \cdot \mu \cdot [P] \cdot k_{on}^{diff}} + k_{cat}} \tag{13}$$

In *E. coli*, the molecular weight of the ribosome is 2307.0 kDa and the molecular weight of a TC is 69.6 kDa [39], thus $a = 33.1$. For a single TC, $K_{m\text{-singleTC}} = 3$ μM [5]; the effective number of TC [5] is 34 (the predicted expressed tRNA in Ref. [39]), and thus $K_m = 34 \cdot K_{m\text{-singleTC}} =$

102 μM. $k_{\text{cat}} = 22 \text{ s}^{-1}$ is the observed maximal translation rate of a ribosome [5], and $k_{\text{on}}^{\text{diff}} \approx k_{\text{cat}}/K_{\text{m}} = 0.216 \text{ μM}^{-1}\text{s}^{-1}$.

The protein concentration [P] is calculated from *E. coli* proteome expression data [49] and cell volume [63] for growth on glucose,

$$[P] = \frac{\sum_i N_i L_i}{V_{\text{cell}} N_{\text{A}}}, \tag{14}$$

where $N_i$ is the copy number per cell and $L_i$ the length of protein *i* [49], $V_{\text{cell}}$ is the cell volume [63], and $N_{\text{A}}$ is the Avogadro constant. In a more recent publication [64], the authors of Ref. [63] re-measured the volume of cells by super-resolution microscopy and found that cell volume was overestimated in Ref. [63] by a factor of $0.67^{-1}$ for growth on glucose. We thus modified cell volume by a factor of 0.67 relative to the values in Ref. [63], resulting in [P] = $1.16 \times 10^6$ μM.

By multiplying the left-hand side of Eq (13) with the molecular weight ratio of tRNA to rRNA, we obtain the tRNA and rRNA mass ratio (Eq (2) of the main text),

$$\frac{M_{\text{tRNA}}}{M_{\text{rRNA}}} = \frac{[TC] \cdot m_{\text{tRNA}}}{[R] \cdot m_{rRNA}} = r \cdot \frac{k_{\text{cat}}}{\sqrt{a \cdot \mu \cdot [P] \cdot k_{\text{on}}^{\text{diff}}} + k_{\text{cat}}}, \tag{15}$$

with

$$r = a \cdot \frac{m_{\text{tRNA}}}{m_{rRNA}} \tag{16}$$

Here, $m_{\text{tRNA}}$ is the molecular mass of tRNA, $m_{\text{rRNA}}$ is the total mass of RNA in one ribosome, and *r* is the ratio of the tRNA mass fraction of a TC and the rRNA mass fraction of the ribosome. For bacteria, we use data from *E. coli* ($m_{\text{tRNA}} = 25.8$ *kDa*, $m_{\text{rRNA}} = 1480$ *kDa*), resulting in *a* = 33.1 and *r* = 0.58. For eukaryotes, we use data from *S. cerevisiae*, resulting in *a* = 40.3 and *r* = 0.59; the molecular weights of the ribosome (3044.4 kDa), rRNA (1750 kDa), TC (75.6 kDa), and tRNA (25.6 kDa) were calculated from the respective sequences according to the *Saccharomyces* Genome Database [65].

## Gene positions

The chromosomal position of the center of the origin of replication (oriC) for different genomes was obtained from the DoriC database (version 10.0) [66]. The start and end positions of rRNA and tRNA genes were downloaded from the RefSeq database (Release 93, downloaded on April 09, 2019); gene locations were defined as the midpoint between gene start and end. We defined gene position as the relative distance of a gene to oriC, calculated as the shortest distance between the gene and oriC on the circular chromosome, divided by half the length of the chromosome. Gene position ranges from 0 to 1.

## Maximal growth rate dataset

Minimal doubling times $\tau_{\text{min}}$ (in hours) were obtained from Ref. [57] and were converted to maximal growth rates as $\mu_{\text{max}} = \frac{\ln(2)}{\tau_{\text{min}}}$. For the analyses, we only used species for which we additionally had genome annotation and oriC location, and which had only one chromosome. The final trimmed dataset contains 170 species (S4 Table).

For 35 out of the 170 species, more than one oriC has been annotated [66]. However, we found that all oriCs are very close on the chromosome in these species: the maximal distance between two oriCs is much less than 1% of the chromosome length (the maximal distance

between two oriCs is 0.0035, equal to 0.18% of the chromosome length). Thus, different oriCs are expected to have a negligible effect on gene position and we randomly selected one of the oriCs to calculate gene position.

## Phylogenetically independent contrasts

16S rDNA sequences was aligned with MUSCLE [67] embedded in MEGA X [68]. A phylogenetic tree was built using maximum likelihood methods with MEGA X with default parameters [68]. The phylogenetic tree was rooted by the minimal ancestor deviation method [69]. We calculated phylogenetically independent contrasts [58] with the pic function in ape package [70] in R [71]. To control for phylogenetic non-independence between data points for different species, we then performed statistical tests on these contrasts ($P_{ic}$ values).

## Gene dosage

We used the Cooper-Helmstetter model [31,32] to calculate gene dosage. The model is briefly summarized below. Let $C$ be the time required to replicate the chromosome; let $D$ be the time between the termination of a round of replication and the next cell division; let $\tau$ be the doubling time. The average dosage of gene $i$ ($\overline{X_i}$) per cell is then given by:

$$\overline{X_i} = 2^{\frac{C(1-position_i)+D}{\tau}}, \tag{17}$$

where $position_i$ is the genomic position of gene $i$. With

$$\tau = \frac{\ln(2)}{\mu}, \tag{18}$$

$$\overline{X_i} = e^{\mu[C(1-position_i)+D]} \tag{19}$$

The gene dosage ratio of two genes ($\overline{X_i}/\overline{X_j}$) is then (Eq (3) of the main text)

$$\frac{\overline{X_i}}{\overline{X_j}} = e^{\mu C(position_j - position_i)} \tag{20}$$

Each genome contains multiple tRNA and rRNA genes. In this case, we use the ratio of the total gene dosages,

$$\frac{\sum \overline{X}_{tRNA}}{\sum \overline{X}_{ribosome}} = \frac{\sum \overline{X}_{tRNA}}{\frac{1}{n}\sum \overline{X}_{rRNA}} = \frac{\sum e^{\mu[C(1-position_{tRNA})+D]}}{\frac{1}{n}\sum e^{\mu[C(1-position_{rRNA})+D]}} = \frac{\sum e^{\mu C(1-position_{tRNA})}}{\frac{1}{n}\sum e^{\mu C(1-position_{rRNA})}}, \tag{21}$$

where $n$ is the number of rRNA genes per ribosome. Since one ribosome contains three rRNA genes (5S, 16S, and 23S rRNA), n = 3.

We assumed a constant DNA replication rate of $k_{rep}$ = 1000 bp s$^{-1}$ [29] to calculate the C-period as

$$C = \frac{L_{genome}}{2k_{rep}}, \tag{22}$$

with $L_{genome}$ the length of the given genome.

## Supporting information

**S1 File. Supplementary file, including supplementary texts and figures.**
(PDF)

**S1 Table. Ternary complex per ribosome in *E. coli* (source data).**
(XLSX)

**S2 Table. tRNA per rRNA in other species (source data).**
(XLSX)

**S3 Table. Fitted ribosome turnover number ($k_{cat}$) by Eq (1) and effective ribosome turnover number ($k_{eff}$) for species in Fig 2 (source data).**
(XLSX)

**S4 Table. The maximal growth rate dataset, including tRNA and rRNA positions, copies, and dosages (source data).**
(XLSX)

**S5 Table. Species-specific replication rate (source data).**
(XLSX)

## Acknowledgments

We thank Mayo Röttger for help with the phylogenetic analysis. We thank Hugo Dourado, Peter Schubert, and Deniz Sezer for helpful discussions.

## Author Contributions

**Conceptualization:** Xiao-Pan Hu.

**Data curation:** Xiao-Pan Hu.

**Formal analysis:** Xiao-Pan Hu.

**Funding acquisition:** Martin J. Lercher.

**Investigation:** Xiao-Pan Hu.

**Methodology:** Xiao-Pan Hu.

**Software:** Xiao-Pan Hu.

**Supervision:** Martin J. Lercher.

**Validation:** Xiao-Pan Hu.

**Visualization:** Xiao-Pan Hu.

**Writing – original draft:** Xiao-Pan Hu.

**Writing – review & editing:** Xiao-Pan Hu, Martin J. Lercher.

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
