## [Decision Letter · Decision Letter 0]

3 Apr 2021

Dear Dr Lercher,

Thank you very much for submitting your Research Article entitled 'An optimal growth law for RNA composition and its partial implementation through ribosomal and tRNA gene locations in bacterial genomes' to PLOS Genetics.

The manuscript was fully evaluated at the editorial level and by independent peer reviewers. The reviewers appreciated the attention to an important problem, but raised some substantial concerns about the current manuscript. Based on the reviews, we will not be able to accept this version of the manuscript, but we would be willing to review a much-revised version. We cannot, of course, promise publication at that time.

If you decide to revise the manuscript for further consideration at PLOS Genetics, please aim to resubmit within the next 60 days, unless it will take extra time to address the concerns of the reviewers, in which case we would appreciate an expected resubmission date by email to plosgenetics@plos.org.

[LINK]

We are sorry that we cannot be more positive about your manuscript at this stage. Please do not hesitate to contact us if you have any concerns or questions.

Yours sincerely,

Eduardo P. C. Rocha

Guest Editor

PLOS Genetics

Josep Casadesús

Section Editor: Prokaryotic Genetics

PLOS Genetics

Thank your for submitting your work to PLoS Genetics. Three reviewers have now evaluated your manuscript. All judged the work interesting and worthy. Yet, they all have substantial criticisms/suggestions that must be addressed in a revision before we can consider the manuscript for publication. Among them, we would like to emphasize the relevance of the following ones:

- Need to match the adequacy of the claims to the data that is presented.

- Discussing, or even better testing, the relevance of the model in the light of previous models.

- Better addressing some biological points, notably showing the separate analysis of tRNA and rRNA (and eventually tRNA per translating ribosomes), the effect of genome size and of temperature. Some of this information may not reveal very significant results, but will answer to questions raised by the study (and part could be put in supplementary material).

- Quantify or discuss the limitations of the model.

- Improve the quantification of the fit between model and data and add some more statistics regarding tests of hypotheses.

Reviewer's Responses to Questions

**Comments to the Authors:**

Reviewer #1: The study suggests a strong relation between tRNA/rRNA ratio with growth rate. They suggest that this relation is under selection via the fact that rRNA genes are typically closer to the origin of replication than tRNA genes and thus have increasingly higher gene dosage at faster growth. The study is potentially interesting but as explained below there are various points that should be improved and it is hard for me to interpret the results without this additional info.

1) abstract: “At the highest growth rates in E. coli, the tRNA/rRNA ratio appears to be regulated entirely through this effect.” I do not see a support to this claim in the paper. The highest correlation reported in the paper is lower than 0.6 (related to 36% explained varinace).

2) It will be interesting to show that the observed signal is correlated with selection pressure (e.g. effective population size).

3) The paper focus on the tRNA/rRNA ratio. Does this ratio correlated with the number of tRNA genes and/or the number of rRNA genes in the genome?

4) How is the reported relation related to the length of the genome.

5) “we here analyze a coarse-grained translation model that only

considers peptide elongation, where the active ribosome acts as an enzyme that converts ternary complexes (TC), consisting of elongation factor Tu (EF-Tu), GTP, and charged tRNA, into an elongating peptide chain following Michaelis-Menten kinetics ..” This model includes many extreme approximations of the translation process (e.g. it does not include traffic jams of ribosomes, their movements, number of ribosomes and tRNAs in the cell, etc). You should evaluate the bias due to these approximations and show that they do not bias the conclusions.

6) " For six out of the seven datasets in Fig. 1, the tRNA/ribosome ratio decreases with increasing growth rate. The only exception, the cyanobacterium Synechococcus elongatus, has a much smaller maximal growth rate (μmax=0.23h-1) than the other species, and its tRNA/rRNA ratio does not show a clear growth ratedependence

[48]. "

“Moreover, a strong selection pressure towards optimal tRNA/ribosome ratios appears

to exist in fast-growing bacteria (Fig 1b).”

All claims should be followed by p-values. This is a general comment related to all results in the paper.

7) Figure 1: the theoretical graphs are not so close to the empirical points. You should at least report a quantitative measure relating to the fit of the theoretical graphs the empirical points and prove that it is significant.

8) The reported p-values (related to the correlations) assume that the points are independent. However, this is not the case for organisms (that were evolve from a common ancestor). You should correct for this when you compute p-values (there are approaches that consider the evolutionary tree when computing the correlations and p-values).

9) It will be helpful to see all the information related to the distributions of distances of of rRNA genes and tRNA from origin of replications in the analysed organisms.

10) “Moreover, we found that both rRNA and tRNA

genes tend to be located ever closer to oriC with increasing μmax (Spearman’s rank correlation coefficient between μmax and position rRNA: = −0.59, = 9.2 × 10/1; between μmax and position tRNA: = −0.40, = 0.0047). ..” please provide all the dot plots related to the correlations. Do you control for the differences in the size of the genomes among the analysed organisms and the number of oriC?

11) Your model assumes that all the tRNA are identical while eventually they decode codons with different frequencies in the transcriptome and AA with different frequencies in the proteome. Do you see differences in terms of the distances of the different tRNA from oriC?

12( how do you decide on the treshold of slow and high growth rate ?

13) “While slowly growing species show a wide range of tRNA/ribosome gene dosage ratios, the ratio in fast-growing species shows a much tighter distribution.” This may be related to tighter selection for fast growing bacteria related to other (e.g. genome size, tRNA copy numbers, etc )

Reviewer #2: In this work, the Authors propose an optimality principle for the expression of ribosomes and associated proteins across organisms, and investigate the role of gene concentration in setting the correct tRNA/rRNA ratio. I was surprised by the short length of the manuscript, but the Authors were able to make their points clearly and concisely. The manuscript could be improved upon by providing some additional comments and analysis, which I list below.

About the optimality principle: how do the predictions compare to other proposed explanation of the tRNA/ribosome ratio, e.g. based on molecular crowding (Klumpp et al., PNAS 2013)? The proposed principle would gain a lot of credibility if alternative hypothesis did a worse job at explaining the data.

Already in the abstract, the Authors state that "at the highest growth rates in E. coli, the tRNA/rRNA ratio appears to be regulated entirely" from the effect of varying gene dose across growth rates. However, the predicted effect is quite small (Fig. 1a): changing growth rate from 1/h to 2/h, the blue line changes by about 10%. The available experimental data seems too scattered to be able to confirm such tiny effect; the Authors should instead state that the data and the predition are compatible, i.e. there is no necessity of additional regulation (although it cannot be excluded).

The Authors did not comment at all about the values of k fitted in Fig. 1b. Did you obtain any biological insight from the observed values of k? To validate the model proposed by the Authors, it would be interesting to compare the values of k=sqrt(k_cat K_M) fitted for the various organisms in Fig. 1b to values obtained from experimentally measured k_cat and K_M at least for some species. Such comparison would considerably strengthen the optimality principle argumented by the Authors.

Here below I have more technical comments:

- Fig. 2e: The value of the C period is hardly known for most cells. The Authors make use of a reasonable estimate of C obtained dividing the size of the chromosome by a fixed replication rate k_rep. However, even Ref. [27] indicates that slow-growing organisms can have much slower replication rates. What would change in Fig. 2e if the k_rep increased with the maximum growth rate of the organisms varied across organisms?

- In the derivation of Eq. (2), the authors equate the protein synthesis flux to the product of the growth rate and the protein concentration. While this should be mostly fine for fast growing cells, I wonder what impact protein degradation has on the results. How would the fitted lines change by assuming a nonzero protein degradation rate?

Another minor comment for Fig. 2ef: I wonder if results would be better shown in double-log scale (including the lack of correlation between tRNA/chromosome and mu_max for fast growing cells). The authors can easily provide a supplementary figure.

Reviewer #3: I performed this review with two close collaborators on this topic.

In this study, Hu and Lercher describe a previously overlooked growth law linking the ratio of tRNA to rRNA to growth rate. After deriving the law (Eq. 1) from a optimality hypothesis and based on a previous (but unpublished) result (ref. 9), they show that the law holds using data from E. coli (Fig 1a) and across microbes (Fig 1b). Then they use the Cooper-Helmstetter model to show that a prediction of tRNA to rRNA ratio based on gene dosage reproduces the optimal prediction (quantitatively at fast growth) across conditions and species (Fig 1b and Fig 2).

The reasoning is convincing, the results appear to be quite solid and our overall impression of the study is positive, although the text is rather condensed and it could be clearer. Definitely this study fits the standards of the journal and it would make an interesting read for anyone interested in cell growth and its impact across genomes.

We propose below some revision points that in our opinion could make the study stronger and easier to read.

%%%%%% Major Comments / Questions %%%%%%%

- Should this result justify the introduction of a concept of “RNA sectors” and their partitioning?

- The authors mention that other studies on growth laws assumed the tRNA to rRNA ratio to be constant. It would be really interesting to know whether the growth law they find has an impact on other growth laws that were derived based on the constancy assumption, and whether the discrepancy can be observed in some data (for example data from Dai et al 2016).

- The authors are basically looking at how tRNAs are repartitioned among available ribosomes (tRNA/rRNA). What will happen if we compare tRNA per *translating* ribosome? Should it be a constant?

- Some central results rely on previous results by the same group, but the text should be made more self-contained. In particular:

- Eqs [4] and [5], which give Eq [1] come almost directly from ref. 9. It would make the life of a reader much easier if they were at discussed and motivated a bit more, at the cost of a small overlap with ref 9 (oddly this is only a preprint although quite old)

- In the current text the hypothesis of optimality is not sufficiently discussed and clarified, both in the results and in the discussion. Again, it is clear to us that this is part of past work, but in our view the extra explanations are required to make this study a bit more self-contained.

A big caveat on the dosage results is that (at least for E. coli), the C period has a strong dependency on temperature (we are modelers ourselves, but we believe it could be a factor of 3 from 37 to 28 degrees), which is a source of variability in the data. This point should be raised by the authors. It can also be addressed in the model, at least of E. coli, looking at the predictions vs temperature and possibly also going back to data from experiments performed at different temperatures.

%%%%%% More specific comments to improve clarity and communication %%%%%%

Can the authors clarify how *"the observation that the cellular dry mass density is approximately constant across growth conditions"* leads to the hypothesis of a *" minimization of the total mass of all participating molecules "*.

Figure 1. It seems the authors should detail how they computed the solid and dashed blue lines. In panel b, model does not seem to do an excellent job, at least looking how the data are presented now, maybe the discrepancies should be discussed.

Connected: with the data in Fig 1b one could try a data collapse based on Eq [1], to show that the data are compatible with a “universal” law across species. For example one could easily collapse by r, and show that two clusters (plus synechococcus) appear, then extract a and look at the values (but possibly something much smarter can be done…)

Line 103. The discussion about gene dosage appears to start out of the blue. It is however linked with what said before. Can the authors improve the transition for the reader?

Line 110. *"Here and below, for each genome, we summarize the multiple tRNA genes by averaging over their positions; we do the same for the rRNA genes."* I can see how that simplifies the analysis, but the risk is to wash out important gene-dependent regulation. Codon bias, etc etc. Maybe for a future work?

In Eq(16) we did not understand why there is $n$ for the ribosomes but not the equivalent for tRNAs.

Line ~135. *"Here, the tRNA/rRNA dosage ratio at higher growth rates (1h/0 ≤ $\\mu$ ≤ 2h/0 ) is very close to the optimal prediction, which corresponds to about 9 tRNAs per ribosome. "* I might have lost it, but from where?

Just after, there is something about the lines in Fig.1, which are actually not explained.

%%%%%% Additional Minor Comments %%%%%%

A sketch or cartoon of the Michaelis-Menten model used to derive Eq [1] could be useful to help the reader.

In our view Fig 1a is arguably the most important of this study. It could be given more graphical emphasis / graphical explanation.

The authors use TC and not tRNA data. This includes elongation factor Tu (EF-Tu), GTP, and _charged_ tRNA (so it depends on aa). Is this a problem?

Is it possible to make a back of the envelop estimate of $\\mu_{max}$ from the position of rRNA and tRNA genes?

Fig.2 The choice of $\\mu_{max}$ to distinguish slow fast is in principle arbitrary, but from (for instance) tRNA position it is evident. Is there something behind that?

The result on gene dosage appears across Fig 1b and 2. Maybe a more optimal figure order can be attempted.

**Have all data underlying the figures and results presented in the manuscript been provided?**

Reviewer #1: **No: **see comments to the authors

Reviewer #2: Yes

Reviewer #3: Yes

PLOS authors have the option to publish the peer review history of their article (what does this mean?). If published, this will include your full peer review and any attached files.

Reviewer #1: No

Reviewer #2: No

Reviewer #3: No

---

## [Decision Letter · Decision Letter 1]

14 Oct 2021

Dear Dr Lercher,

Thank you very much for submitting your Research Article entitled 'An optimal growth law for RNA composition and its partial implementation through ribosomal and tRNA gene locations in bacterial genomes' to PLOS Genetics.

The manuscript was fully evaluated at the editorial level and by independent peer reviewers. The reviewers appreciated the attention to an important topic but identified some concerns that we ask you address in a revised manuscript

We therefore ask you to modify the manuscript according to the review recommendations. Your revisions should address the specific points made by each reviewer.

[LINK]

Yours sincerely,

Eduardo P. C. Rocha

Guest Editor

PLOS Genetics

Josep Casadesús

Section Editor: Prokaryotic Genetics

PLOS Genetics

The reviewers are very happy with the changes you introduced in the manuscript and the editors are equally satisfied. One of the reviewers has some minor comments that you might find relevant and this is why this is marked as a minor revision. Thank you for submitting this very interesting work to PLoS Genetics.

Reviewer's Responses to Questions

**Comments to the Authors:**

Reviewer #2: By addressing my comments and the comments from the other reviewers, the Authors have greatly improved the manuscript. I think that the revised manuscript adheres to the high standards of quality and broad relevance of PLOS Genetics, and I am happy to recommend its publication.

Reviewer #3: I am happy with the authors’ replies and revisions.

Some minor comments that the authors might optionally consider for a brief discussion

In this recent study, the authors derive a tRNA growth law (Eqs 6-9), which might be interesting to compare with this study

PMID: 34389683

One additional caveat that may be mentioned is the physiological dependency of the C period on growth rate

PMID: 32424336

PMID: 12686642

There are two other attempts (from the same group) of cross species growth law that the authors might want to compare with their approach

PMID: 27046336

PMID: 22203990

Concerning the role of degradation (asked by another referee), there is a preprint

https://doi.org/10.1101/2021.03.25.436692

**Have all data underlying the figures and results presented in the manuscript been provided?**

Reviewer #2: Yes

Reviewer #3: Yes

PLOS authors have the option to publish the peer review history of their article (what does this mean?). If published, this will include your full peer review and any attached files.

Reviewer #2: No

Reviewer #3: No

---

## [Editor Report · Decision Letter 2]

10 Nov 2021

Dear Dr Lercher,

We are pleased to inform you that your manuscript entitled "An optimal growth law for RNA composition and its partial implementation through ribosomal and tRNA gene locations in bacterial genomes" has been editorially accepted for publication in PLOS Genetics. Congratulations!

Yours sincerely,

Eduardo P. C. Rocha

Guest Editor

PLOS Genetics

Josep Casadesús

Section Editor: Prokaryotic Genetics

PLOS Genetics

Comments from the reviewers (if applicable):

Thank you for considering the novel comments by the reviewers. It is now our great pleasure to accept your excellent work for publication in PLoS Genetics.

**Data Deposition**

http://datadryad.org/submit?journalID=pgenetics&manu=PGENETICS-D-21-00268R2

**Press Queries**

---

## [Editor Report · Acceptance letter]

22 Nov 2021

PGENETICS-D-21-00268R2 

An optimal growth law for RNA composition and its partial implementation through ribosomal and tRNA gene locations in bacterial genomes 

Dear Dr Lercher, 

We are pleased to inform you that your manuscript entitled "An optimal growth law for RNA composition and its partial implementation through ribosomal and tRNA gene locations in bacterial genomes" has been formally accepted for publication in PLOS Genetics! Your manuscript is now with our production department and you will be notified of the publication date in due course.

With kind regards,

Livia Horvath

PLOS Genetics

On behalf of:
